# Impact of Hepatitis B Virus Infection, Non-alcoholic Fatty Liver Disease, and Hepatitis C Virus Co-infection on Liver-Related Death among People Tested for Hepatitis B Virus in British Columbia: Results from a Large Longitudinal Population-Based Cohort Study

**DOI:** 10.3390/v14112579

**Published:** 2022-11-21

**Authors:** Jean Damascene Makuza, Dahn Jeong, Mawuena Binka, Prince Asumadu Adu, Georgine Cua, Amanda Yu, Héctor Alexander Velásquez García, Maria Alvarez, Stanley Wong, Sofia Bartlett, Mohammad Ehsanul Karim, Eric M. Yoshida, Alnoor Ramji, Mel Krajden, Naveed Zafar Janjua

**Affiliations:** 1School of Population and Public Health, The University of British Columbia, Vancouver, BC V6T 1Z3, Canada; 2Clinical Prevention Services, British Columbia Centre for Disease Control, Vancouver, BC V5Z 4R4, Canada; 3Centre for Health Evaluation and Outcome Sciences, St Paul’s Hospital, Vancouver, BC V6Z IY6, Canada; 4Division of Gastroenterology, The University of British Columbia, Vancouver, BC V5Z 1M9, Canada

**Keywords:** hepatitis B virus, liver-related mortality, British Columbia, population-based cohort

## Abstract

Data on the contribution of hepatitis B virus (HBV) infection and related comorbidities to liver-related mortality in Canada are limited. We assessed the concurrent impact of HBV infection, non-alcoholic fatty liver disease (NAFLD), and hepatitis C virus (HCV) coinfection on liver-related deaths in British Columbia (BC), Canada. We used data from the BC Hepatitis Testers Cohort (BC-HTC). We used Fine–Gray multivariable sub-distributional hazards models to assess the effect of HBV, NAFLD, and HCV coinfection on liver-related mortality, while adjusting for confounders and competing mortality risks. The liver-related mortality rate was higher among people with HBV infection than those without (2.57 per 1000 PYs (95%CI: 2.46, 2.69) vs. 0.62 per 1000 PYs (95%CI: 0.61, 0.64), respectively). Compared with the HBV negative groups, HBV infection was associated with increased liver-related mortality risk in almost all of the subgroups: HBV mono-infection (adjusted subdistribution hazards ratio (asHR) of 3.35, 95% CI 3.16, 3.55), NAFLD with HBV infection, (asHR 12.5, 95% CI 7.08, 22.07), and HBV/HCV coinfection (asHR 8.4, 95% CI 7.62, 9.26). HBV infection is associated with a higher risk of liver-related mortality, and has a greater relative impact on people with NAFLD and those with HCV coinfection. The diagnosis and treatment of viral and fatty liver disease are required to mitigate liver-related morbidity and mortality.

## 1. Introduction

The World Health Organization (WHO) estimated that 296 million people were living with chronic hepatitis B virus (HBV) infection worldwide as of 2020 [1]. It was further reported that 820,000 people died of HBV-related causes in 2019 [1]. HBV-related cirrhosis and hepatocellular carcinoma are associated with a high morbidity and mortality [2]. Liver-related mortality in Canada has been increasing and is predicted to continue to rise due to hepatitis C virus (HCV) infection, as well as alcohol and non-alcohol-related liver diseases [3,4,5,6,7]. However, data on the contribution of HBV infection to liver-related mortality in Canada are scarce.

In 2015, approximately 285,000 Canadians were living with HBV [8]. Immigrants from East Asia (5–15%) had a higher prevalence than the Canadian-born population (0.24–0.5%) [9]. In British Columbia (BC), among the people diagnosed with HBV infection, 4.3% had acute HBV infection, while 95.7% had a chronic infection [10]. In addition, acute HBV infection was predominantly diagnosed among White people (78%), while 60% of chronic HBV infections were among people of an East Asian origin [10]. Populations with a higher prevalence of HBV infection include people from East Asia; gay, bisexual, and other men who have sex with men (gbMSM); and people who use injection drugs (PWID) [11,12,13]. Recent studies in BC showed that HBV infection is associated with a high morbidity and mortality [11].

Previous studies have reported that non-alcoholic fatty liver disease (NAFLD) and HBV/HCV coinfection are associated with a higher risk of liver cirrhosis and hepatocellular carcinoma [14,15]. The findings from a study conducted in the United States showed that in 2020, people with chronic HBV infection had a 13.3 fold increased risk of dying from liver-related mortality compared with people with no HBV infection [16]. People with chronic HCV infection are at higher risk of dying from liver-related death compared with people without HCV infection [17]. The treatment of HCV infection with direct-acting antiviral agents (DAAs) is associated with a reduction in liver-related mortality [6,18]. Several studies have shown that the risk of liver-related mortality is greater for individuals with coinfection of HBV and HCV than mono-infected patients [15,19,20]. The literature has shown a high risk of liver-related mortality among people with NAFLD compared with the general population [21]. However, to the best of our knowledge, no study has assessed the relationship between HBV infection, concurrent NAFLD and HCV co-infection, and liver-related mortality, each of which are independently associated with a higher mortality. In this longitudinal cohort study, we estimated the liver-related mortality rate and the concurrent impact of HBV infection, NAFLD, and HCV coinfection on liver-related deaths in BC using population-based data from the BC Hepatitis Testers Cohort (BC-HTC) [22]. 

## 2. Patients and Methods 

### 2.1. Data Source, Design, and Study Population

This analysis is based on the BC-HTC, which includes more than 1.7 million individuals tested for HCV or HIV, or reported as a case of HCV, HIV, or HBV from 1990 to 31 December 2015. These data are integrated through a personal health number with various administrative healthcare datasets and registries including vital statistics, cancer registries, medical visits, hospital discharge data, emergency visits, and drug dispensations [22]. A more detailed description of BC-HTC has been reported previously [12,13,22]. 

The current analysis included 1.1 million individuals who were ever tested for HBV as of 30 June 2015. We excluded people with missing data or an undefined response on outcome and those with unknown information regarding sex (Figure 1). This study was approved by the University of British Columbia Behavioral Research Ethics Board (H14-01649).

### 2.2. Outcome and Exposure

The outcome of this study is liver-related death. Assessment of liver-related deaths was based on the BC Vital Statistics Registry, including data up to 30 June 2020 [6]. Liver-related death was defined using the International Classification of Diseases, tenth (ICD-10) edition [23,24]. The underlying and contributory cause of death were both used for defining the causes of death. People with any of the following ICD-9 code: 571 (for the period 1990–2007) and ICD-10 codes (for the period 2008–2020): viral hepatitis (B15-19), sequelae of viral hepatitis (B942), liver cancer (C22), alcoholic liver disease (K70), and non-alcoholic liver disease (K71-77) were considered as liver-related deaths. Deaths from causes other than liver-related cause were considered as competing risks [25].

The exposure variable for this study was HBV test positivity, based on the BC viral hepatitis testing guidelines [26]. In BC HTC, an individual was confirmed as a case of HBV infection with a positive HBV DNA or HBVe antigen (HBeAg) test result, or those who had a record of receiving HBV treatment [10]. The HBV effect on mortality was extended to include the concurrent effect of HCV and NAFLD. NAFLD was defined by at least one physician billing or a hospitalization code related to another chronic non-alcoholic liver disease (IDC-9 code 571.8), other specified inflammatory liver diseases (ICD-10 code K-75.8), or fatty (change of) liver, not elsewhere classified (ICD-10 code K-71.0, tenth edition) [15,16,19]. Chronic HCV infections were diagnosed based on laboratory confirmation according to BC provincial guidelines for HCV testing from a positive HCV RNA test or based on treatment data [26].

### 2.3. Potential Risk Factors

Demographic variables, including age (categorized as <25, 25–34, 35–44, 45–54, 55–64, and 65 and above), sex, and ethnicity were identified as risk factors for HBV [27,28]. Ethnicity was determined with validated name recognition programs [29,30], and was classified into three categories (South Asian, East Asian, and Other (White, Black, Central Asian, Latin American, Pacific Islander, and West Asian individuals)). Socioeconomic status was also assessed using the Québec Index of Material and Social Deprivation, defined in five quintiles from Q1 to Q5 [31]. The material component consisted of indicators for education, employment, and income (persons without high school diplomas, ratio employment–population ratio, and average personal income). The social component comprised indicators related to marital status and family structure (persons living alone, persons separated, divorced or widowed, and single-parent families) [11]. We defined history of injection drug use (IDU), gbMSM, and problematic alcohol use at the time of the last HBV diagnosis or the last negative test (for negative individuals) using validated algorithms, and determined the occurrence of either one hospitalization or ambulatory care visit, or two physician visits pertaining to these risk factors [32,33]. Having other comorbidities such as diabetes, chronic kidney disease, hypertension, and hepatic complications liver cirrhosis were assessed using validated algorithms based on diagnostic codes and/or prescription drug records in administrative health datasets [33]. HIV infections were diagnosed based on laboratory confirmation according to BC provincial guidelines for HIV testing [26,34].

### 2.4. Statistical Analysis

The primary analysis for this study included the entire study population. We assessed the crude liver-related mortality rate by HBV infection status. Person-time started at the HBV positive test date for HBV positive individuals and the last HBV test date for negative individuals, and ended at death or end of follow-up (30 June 2020).

In this study, the distribution of characteristics in individuals who tested positive for HBV infection compared with those who tested negative was different. We compared the characteristics of those who tested negative and those who tested positive through standardized mean differences (SMD) [35]. We considered the SMD cut-off point at 0.2. To account for differences between individuals who tested positive and those who tested negative, we estimated the propensity scores (PS) for infection through a logistic regression model. The model included sex, ethnicity, material deprivation quintiles, HCV infection, HIV infection, diabetes, chronic kidney disease, injection drug use, alcohol use disorder, and non-alcoholic fatty liver disease. The criteria for covariate selection for the PS model was whether the covariates confounded the relationship between HBV and liver-related mortality or risk factors of liver-related mortality [36], in order to avoid an imbalance between variables. The inverse probability of the treatment weighting (IPTW) estimating average treatment effect (ATE) based on PS was conducted as the sensitivity analysis [37]. In IPTW, we matched each HBV positive individual with a negative individual.

Cumulative incidence curves and incidence rates per 1000 person-years (PY) of follow-up were calculated in the overall study population using the HBV infection status for each NAFLD group and HCV infection group. We created four-level variables that combined HBV infection and NAFLD status, as well as HBV infection and HCV status, to assess and compare the risk of liver-related mortality across the eight subgroups (Non-NAFLD, with NAFLD, Non-HCV, with HCV without HBV infection; Non-NAFLD, with NAFLD, Non-HCV, with HCV with HBV infection). A multivariable Fine–Gray proportional hazards model [38] was used to compute the adjusted sub-distributional hazard ratios (asHR) for liver-related mortality in each subgroup. Finally, we conducted a stratified analysis to compute asHR for the effect of HBV infection on liver-related mortality by NAFLD status and by HCV status. To assess the effect modification of HBV infection by NAFLD status and by HCV status, we included the interaction terms in the multivariable model and performed a likelihood ratio test based on the analysis of deviance [39]. In the sensitivity analysis using the IPTW method, the Fine–Gray models were fit in the overall dataset with the IPTW for ATE. The multivariable Fine–Gray model with IPTW also assessed the interaction term. The dataset was created in the Statistical Analysis System (SAS) [40] and was imported in R, version 4.2 [41], for the analyses.

## 3. Results

### 3.1. Study Participant’s Characteristics

Between 1990 and 2015, 1,151,241 people were tested for HBV in BC. After excluding people without data on sex, exposure, and outcome variables, our analysis was conducted on 1,150,994 individuals (Figure 1). Among these 1,150,994 individuals, 45,255 (4.09%) were HBV positive and 1801 (3.98%) individuals with HBV infection died from liver-related mortality. In total, 8963 (0.81%) individuals died from liver-related mortality among the 1,105,739 individuals without HBV infection. The median follow-up time for the study sample was 11.86 (interquartile range: 7.55–18.01) years. A higher proportion of those testing positive compared with negative were males (25,019 (55.3%) vs. 488,450 (44.2%)), born between 1945 and 1964 (21,107 (46.6%) vs. 331,100 (29.9%)), and of East Asian ethnicity (26,827 (59.3% vs. 148,383 (13.4%)). In the social deprivation categories, a higher proportion of those testing positive compared with negative was found in most deprived people (12,654 (28.0%) vs. 214,970 (19.4%)). Among the material deprivation categories, a higher proportion of those testing positive compared with negative was found in most privileged people (10,326 (22.8%) vs. 202,776 (18.3%)) (Table 1).

IPTW also reduced the differences in characteristics between individuals with HBV infection and those without it, as shown by the SMDs in Table 1.

### 3.2. Liver-Related Mortality Rate among Participants

In the overall population, the highest mortality rate was found in people with HBV infection compared with those without (2.57 per 1000 PY (95% CI: 2.46, 2.69) vs. 0.62 per 1000 PY (95% CI: 0.61, 0.64), respectively). In people with NAFLD, among those without HBV infection, the incidence of liver-related mortality was 6.79/1000 PY (95% CI: 6.66, 6.94) and among people with infection, 27.97/1000 PY (95% CI: 15.88, 49.25). In people without NAFLD, among those without HBV infection, the incidence of liver-related mortality was 0.62/1000 PY (95% CI: 0.61, 0.63) and among people with HBV infection, 2.56/1000 PY (95% CI: 2.44, 2.68). In people with HCV infection, among those without HBV infection, the incidence of liver-related mortality was 5.48/1000 PY (95% CI: 5.17, 5.82) and among people with infection it was 10.39/1000 PY (95% CI: 9.53, 11.34). In people without HCV infection, among those without HBV infection, the incidence of liver-related mortality was 0.56/1000 PY (95% CI: 0.54, 0.57) and among people with infection, 1.99/1000 PY (95% CI: 1.88, 2.10). The Kaplan–Meier survival curves for liver-related mortality showed significantly higher cumulative mortality among people with HBV infection (vs. those without HBV infection) over time, with a much higher mortality among those with concurrent NAFLD or HCV coinfection (Figure 2).

### 3.3. Effect of HBV Infection on Liver-Related Mortality

In the main analysis, after adjusting for the potential confounders and competing mortality risks, HBV infection was associated with a higher risk of liver-related mortality (asHR: 3.34, 95% CI: 3.15, 3.53). These findings from all of the population dataset were similar to those from the PS-weighted ATE (asHR:3.24, 95% CI: 2.99, 3.50) (Table 2). Other factors associated with an increased incidence of liver-related deaths include older age at HBV testing (asHR: 10.32, 95% CI: 8.20, 13.0 for 65 years old and above compared with individuals of 25 years and below), diabetes (asHR: 1.68, 95%CI: 1.59, 1.77), HCV infection (asHR: 3.68, 95% CI: 3.45, 3.93), NAFLD (asHR: 4.31, 95% CI: 3.32, 5.61), and alcohol use disorder (asHR: 3.58, 95% CI: 3.38, 3.80).

### 3.4. The Effect of NAFLD and HCV Infection in the HBV Liver-Related Mortality

In the multivariable model accounting for competing mortality, compared with those without NAFLD and without HBV infection, individuals without NAFLD but with HBV infection had a 3.35 times higher mortality risk (asHR 3.35, 95% CI: 3.16, 3.55). Similarly, individuals with NAFLD without HBV infection had a 4.49 times higher hazard of liver-related mortality (asHR 4.49, 95% CI: 3.35, 6.02), while individuals with both NAFLD and HBV infection had a 12.5 times higher hazard of liver-related mortality (asHR: 12.5, 95% CI: 7.08, 22.07). In the HBV-HCV model, compared with those without HCV infection and without HBV infection, individuals without HCV infection but with HBV infection had a 4.36 times higher risk of mortality (asHR: 4.36, 95% CI: 4.09, 4.64). Individuals with HCV infection without HBV infection had a 4.69 times higher risk of liver-related mortality (asHR: 4.69, 95% CI: 4.37, 5.03), while individuals with both HCV infection and HBV infection had a 8.4 times the risk of liver-related mortality (asHR: 8.4, 95% CI: 7.62, 9.26) compared with those without HCV infection and without HBV infection. The IPW ATE weighted analysis showed a similar asHR (Table 3).

## 4. Discussion

In this population-based Canadian cohort of more than a million individuals, we evaluated the effect of HBV infection and concurrent HCV co-infection and NAFLD on liver-related mortality risk. Individuals with HBV infection had a more than three times higher risk of dying from liver-related death compared with those without HBV infection. In people with both NAFLD and HBV infection, the risk of liver-related mortality was more than 12 times that of those without HBV infection and three times those with HBV or NFLD alone. Similarly, among people with both HBV and HCV coinfection, the risk of liver-related mortality was eight times compared with those without HBV or HCV and two times compared with HBV or HCV alone. These findings highlight that people with HBV infection who have concurrent NAFLD or HCV infection are at a much higher risk of liver-related mortality. Given the availability of highly effective treatments for HCV and HBV, there is a real opportunity to reduce this risk by early treatment for HBV or HCV among these individuals.

Previous studies have shown that the contribution of HBV mono-infection, NAFLD, and HCV alone to increased liver-related alone to mortality [42,43]. However, limited data are available on the concurrent HBV infection and NAFLD and HBV/HCV coinfection and liver-related mortality. Studies from Spain and Taiwan have reported a higher rate of liver-related mortality associated with HBV mono-infected [44,45]. An elevated risk of liver-related mortality among people with NAFLD has been reported from the United States of America (USA) [14]. The findings from the studies conducted in Canada, the Netherlands, and Israel among people with chronic HBV have reported that non-alcoholic steatohepatitis was associated with an increased risk of all-cause mortality [45,46], but their contribution to liver-related death was not assessed. A relatively small study from Spain reported a higher risk of death from liver disease with HCV infection among people with positive HBsAg. In summary, there are limited studies on the effect of HBV infection and NAFLD and HBV and HCV coinfection and liver-related mortality. Our findings suggest that for people with HBV infection, HCV coinfection and NAFLD substantially augment their risk of liver-related mortality.

Studies have reported that liver-related mortality rates among people with HBV mono-infection people vary between 0.1 per 100,000 PYs and 8.6 per 1000 PYs [47,48,49]. In a cohort of 39,109 individuals with HBV infection conducted in Australia from 1990 to 2002, the liver-related mortality among people with HBV infection was lower (1.17 per 1000 PYs) than our results [50]. This could be due to the differences in the periods of study; our study considered people from 1990 to 2020, and so the high mortality in our study could be related to an age difference in the population. In our study, the risk of death was high in older people. In another cohort study also conducted in Australia from 1993 to 2012, 4.8% of HBV positive patients died from liver-related mortality, with an increased rate of liver-related death in older individuals [48]. However, our results are far lower than those found in a study conducted in Italy, where the mortality rate among patients with HBV followed for 25 years was 15.7%; however, this study was conducted on only 70 HBV patients [2].

In addition to HCV coinfection and NALFD, other factors associated with a higher liver-related mortality among people with HBV infection included advanced age and alcohol use disorder. All of these are factors that were shown to contribute on their own to liver-related mortality through several studies in different areas. Other studies have shown the association of aging and alcohol disorder problems with liver-related complications, including death [51].

This study has several strengths. This is the largest population-based study to date to investigate the effect of HBV infection on mortality in BC. In addition, the study design allowed us to establish the temporality in the association of HBV and liver-related mortality. Even though this is an observational study, to avoid residual confounding and other biases, we used a robust approach, which allowed for a less biased assessment of the impact of HBV infection on liver-related mortality by balancing the baseline characteristics between individuals with and without HBV infection [52].

The findings of this study should be appraised in the context of the following limitations. First, in BC, HBV notifications are based on evidence of chronic infection. However, there is a probability of overestimating the impact of HBV on liver-related mortality as this study used data for all people who tested positive for HBV without the distinction of chronic or acute, so some individuals may have cleared HBV infection. Secondly, for people with multiple chronic diseases that are each potentially fatal, the decision about selecting the underlying cause of death can be subjective; however, where possible (i.e., liver-related mortality), a multiple cause definition was used to include important cause information that might have been otherwise overlooked. In addition, using administrative data has limitations because of a lack of sociodemographic and health risk information such as marital status, level of education, smoking, poor diet, and physical inactivity, which could be among the risk factors for liver-related mortality. As unmeasured confounders, these could introduce bias and underestimate the relationship between HBV and liver-related mortality [53]. Finally, the major limitation in this analysis relates to the need to account for treatment to assess the impact of non-cleared HBV. Future studies on liver-related mortality and all-cause mortality should consider impact of treatment. To account for residual confounding we used a double-robust approach of weighting based on PS, and by adjusting for covariates in the proportional hazards. 

## 5. Conclusions

In this large population-based cohort of individuals tested for HBV in BC, we found that HBV infection is associated with a higher risk of liver-related mortality, with a greater relative impact in people with NAFLD and those with HBV/HCV co-infection. This indicates an urgent need for effective HBV and HCV treatment, as well as continued monitoring for liver-related diseases in order to reduce the risk of mortality. In addition, highly effective HBV vaccination prevents infection and hence reduces mortality, and thus must be part of the overall strategy to reduce the impact of HBV on health. The results from this study highlight the need for service integration to improve the overall wellbeing and survival of people with multiple co-occurring risk factors.

## Figures and Tables

**Figure 1 viruses-14-02579-f001:**
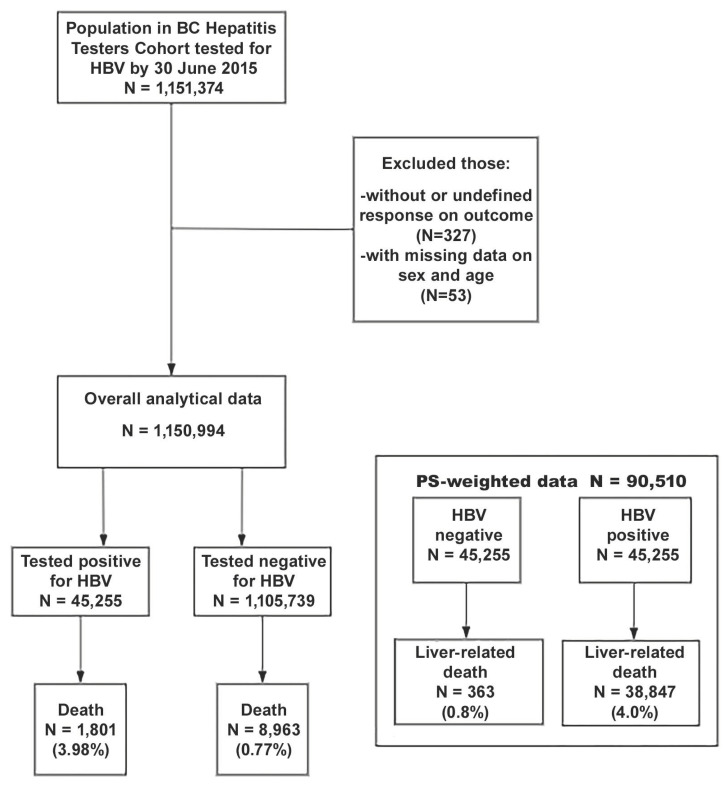
Study population flowchart for liver-related mortality among a BC population cohort tested for HBV from 1990–2015. BC, British Columbia; HBV, hepatitis B virus; PS: propensity score.

**Figure 2 viruses-14-02579-f002:**
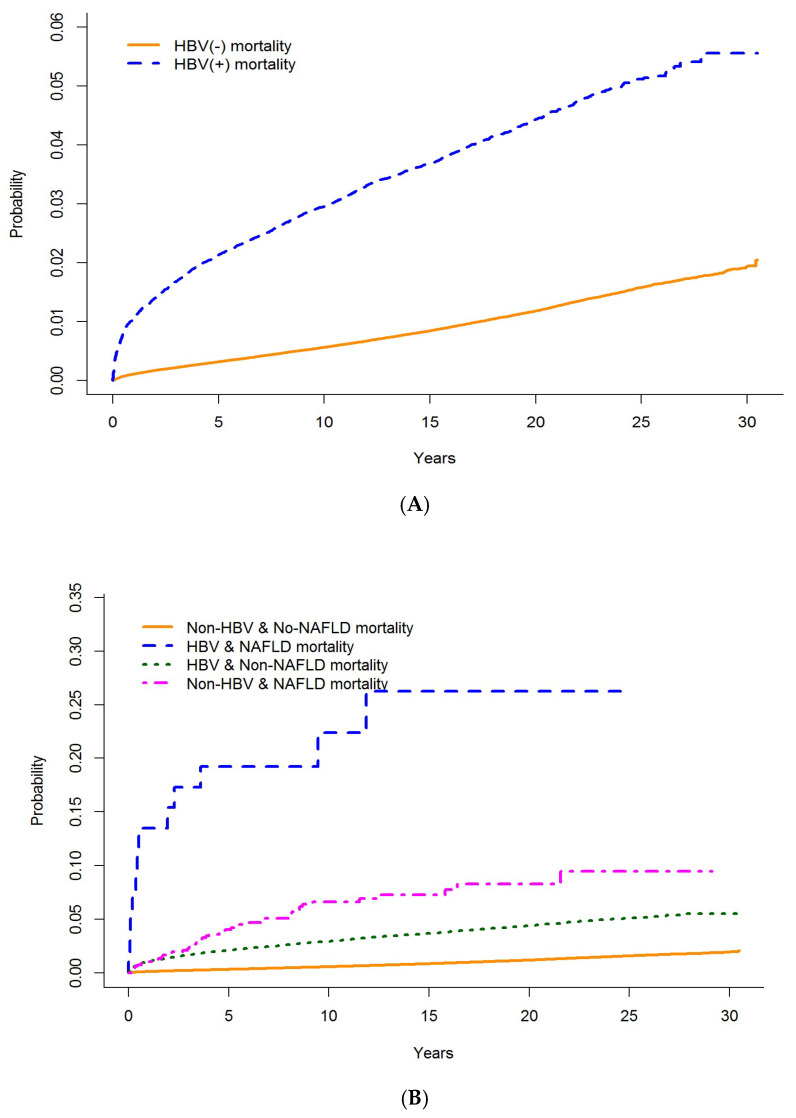
Cumulative mortality rate for people testing HBV positive vs. those testing negative in BC from 1990–2015. (**A**) Cumulative liver-related mortality in individuals with and without HBV infection. (**B**) Cumulative liver-related mortality in individuals with and without HBV infection and co-occurring NAFLD. (**C**) Cumulative liver-related mortality in individuals with and without HBV infection and HCV coinfection.

**Table 1 viruses-14-02579-t001:** Baseline study participants’ characteristics at the first HBV testing by HBV infection status and the propensity score (PS)-weighting dataset from the British Columbia Hepatitis Testers Cohort 1990–2015.

	Overall Study Population			Inverse Probability of Treatment Weighted Dataset *		
Covariates	HBV Negative (n = 1,105,739)	HBV Positive (n = 45,255)	SMD	HBV Negative (n = 45,255)	HBV Positive (n = 45,255)	SMD
Sex (%)			0.224			0.025
Female	657,289 (55.8)	20,236 (44.7)		25,071 (55.4)	24483 (54.1)	
Male	488,450 (44.2)	25,019 (55.3)		20,183 (44.6)	20,772 (45.9)	
Age group, years (%)			0.331			0.033
<25	257,063 (23.2)	6021 (13.3)		10,318 (22.8)	9820 (21.7)	
25–34	301,363 (27.3)	10,634 (23.5)		12,264 (27.1)	12,174 (26.9)	
35–44	215,920 (19.5)	12,128 (26.8)		8960 (19.8)	9413 (20.8)	
45–54	151,346 (13.7)	8804 (19.5)		6290 (13.9)	6336 (14.0)	
55–64	96,763 (8.8)	4677 (10.3)		3982 (8.8)	3439 (9.0)	
≥65	83,284 (7.5)	2991 (6.6)		3394 (7.5)	3439 (7.6)	
Birth group (%)			0.496			0.055
<1945	117,235 (10.6)	5523 (12.2)		4842 (10.7)	5250 (11.6)	
1945–1964	331,100 (29.9)	21,107 (46.6)		13,848 (30.6)	14,572 (32.2)	
1965–1974	217,614 (19.7)	10,183 (22.5)		8960 (19.8)	8870 (19.6)	
≥1975	439,790 (39.8)	8442 (18.7)		17,604 (38.9)	16,563 (36.6)	
Ethnicity (%)			1.189			0.031
Other *	867,680 (78.5)	16,933 (37.4)		34,756 (76.8)	34,213 (75.6)	
East Asian	148,383 (13.4)	26,827 (59.3)		6924 (15.3)	7377 (16.3)	
South Asian	89,676 (8.1)	1495 (3.3)		3575 (7.9)	3666 (8.1)	
Social deprivation (%)			0.276			0.060
Most privileged (Q1)	243,399 (22.0)	6975 (15.4)		8372 (18.5)	8553 (18.9)	
Q2	212,109 (19.2)	6879 (15.2)		8101 (17.9)	8508 (18.8)	
Q3	208,340 (18.8)	8055 (17.8)		7965 (17.6)	7648 (16.9)	
Q4	220,325 (19.9)	9970 (22.0)		9006 (19.9)	8870 (19.6)	
Most deprived (Q5)	214,970 (19.4)	12,654 (28.0)		11,540 (25.5)	11,812 (26.1)	
Unknown	6596 (0.6)	722 (1.6)		272 (0.6)	272 (0.6)	
Material deprivation (%)			0.183			0.024
Most privileged (Q1)	202,776 (18.3)	10,326 (22.8)		9820 (21.7)	8553 (18.9)	
Q2	196,770 (17.8)	9053 (20.0)		8598 (19.0)	8508 (18.8)	
Q3	194,886 (17.6)	7663 (16.9)		8508 (18.8)	8644 (19.1)	
Q4	221,654 (20.0)	7650 (16.9)		9051 (20.0)	9187 (20.3)	
Most deprived (Q5)	283,057 (25.6)	9841 (21.7)		8960 (19.8)	9866 (21.8)	
Unknown	6596 (0.6)	722 (1.6)		272 (0.6)	272 (0.6)	
Hepatitis C virus (yes) (%)	15,906 (1.4)	4139 (9.1)	0.349	2869 (26.9)	1665 (29.9)	0.066
HIV† (yes) (%)	3961 (0.4)	1318 (2.9)	0.202	226 (0.5)	272 (0.6)	0.015
Diabetes (yes) (%)	63,917 (5.8)	2493 (5.5)	0.012	1655 (15.5)	887 (15.9)	0.011
Chronic Kidney Disease (yes) (%)	5342 (0.5)	466 (1.0)	0.063	226 (0.5)	272 (0.6)	0.010
Hypertension (%)	132,663 (12.0)	4403 (9.7)	0.073	5431 (12.0)	4707 (10.4)	0.049
Cirrhosis (yes) (%)	3490 (0.3)	659 (1.5)	0.105	226 (0.5)	226 (0.5)	0.022
Decompensated cirrhosis (yes) (%)	3576 (0.3)	450 (1.0)	0.083	181 (0.4)	181 (0.4)	<0.001
Gay, bisexual and other men who have sex with men (%)	65,394 (5.9)	4,013 (8.9)	0.232	2761 (6.1)	3620 (8.0)	0.082
History of injection drug use (%)	28,814 (2.6)	2318 (5.1)	0.131	1222 (2.7)	1222 (2.7)	0.004
Alcohol use disorder (%)	39,599 (3.6)	1953 (4.3)	0.038	1629 (3.6)	1448 (3.2)	0.022
Non-alcoholic fatty liver disease (%)	668 (0.1)	52 (0.1)	0.018	45 (0.1)	45 (0.1)	0.003

* The propensity scores (PS)-matched dataset used inverse probability weights (IPW) to estimate the average treatment effect (ATE). The PS were computed based on age at the first HBV testing, sex, birth group, ethnicity, social deprivation quintiles, HCV infection, HIV infection, diabetes, chronic kidney disease, history of injection drug use, alcohol use disorder, and non-alcoholic fatty liver disease. The IPW analysis includes 45,255 individuals with HBV infection and 45,255 individuals without HBV infection. HBV, hepatitis B virus; HCV, hepatitis C virus; HIV, human immunodeficiency virus; TB, tuberculosis; PS, propensity scores; IPW: inverse probability weighting; Q, quintile; SD, standard deviation; SMD, standardized mean difference. Other *: White, Black, Central Asian, Latin American, Pacific Islander, and West Asian individuals.

**Table 2 viruses-14-02579-t002:** Impact of HBV infection on liver-related mortality between 1990–2015 in the British Columbia Hepatitis Testers Cohort, using multivariate * analysis Cox regression analysis.

	Overall Study Population		IPW-Dataset ATE **	
Covariates	asHR	95%CI	asHR	95%CI
HBV status				
Negative	Ref		Ref	
Positive	3.34	(3.15, 3.53)	3.24	(2.99, 3.50)
Sex				
Female	Ref			
Male	1.52	(0.21,10.91)	1.99	(1.75, 2.28)
Age, years				
<25	Ref			
25–34	1.46	(1.19, 1.79)	1.34	(0.68, 2.64)
35–44	2.89	(2.33, 3.57)	2.3	(1.18, 4.47)
45–54	4.93	(3.97,6.13)	3.06	(1.57, 5.97)
55–64	7.34	(5.87, 9.17)	4.39	(2.23, 8.66)
≥65	10.32	(8.20, 13.0)	6.08	(3.05, 12.09)
Birth group				
<1945	Ref			
1945–1964	0.68	(0.64, 0.73)	0.49	(0.41, 0.58)
1965–1974	0.27	(0.24, 0.31)	0.23	(0.16, 0.33)
≥1975	0.12	(0.10, 0.15)	0.12	(0.06, 0.24)
Ethnicity				
East Asian	Ref			
South Asian	1.24	(1.11, 1.38)	1.62	(1.46, 1.78)
Other *****	2.03	(1.90, 2.16)	0.93	(0.70, 1.24)
Material deprivation				
Most privileged(Q1)	Ref			
Q2	1.26	(1.18, 1.35)	1.13	(0.92, 1.40)
Q3	1.26	(1.18, 1.35)	1.03	(0.84, 1.27)
Q4	1.34	(1.26, 1.43)	1.21	(0.99, 1.47)
Most deprived(Q5)	1.49	(1.40, 1.57)	1.28	(1.06, 1.54)
unknown	1.46	(1.20, 1.77)	1.3	(0.87, 1.95)
HCV status				
Negative	Ref			
Positive	3.68	(3.45, 3.93)	3.33	(2.96, 3.76)
Diabetes				
No	Ref			
Yes	1.68	(1.59, 1.77)	1.86	(1.58, 2.20)
Chronic Kidney Disease				
No	Ref			
Yes	1	(0.84,1.18)	0.99	(0.67, 1.48)
Injection Drug Use				
No	Ref			
Yes	1.17	(1.06,1.27)	1.12	(0.88, 1.41)
Alcohol use disorder				
No	Ref			
Yes	3.58	(3.38, 3.80)	2.56	(2.13, 3.07)
Non-alcoholic fatty liver disease				
No	Ref			
Yes	4.31	(3.32, 5.61)	3.53	(2.11, 5.91)

* Multivariable models were adjusted with sex, age at first HBV testing date, birth cohort, ethnicity, material deprivation quintiles, HCV infection, diabetes, chronic kidney diseases, alcohol use disorder, injection drug use and other competing risks. asHR, sub-distributional adjusted hazard ratios; CI, confidence interval; HBV, hepatitis B virus. ** The PS-weighted dataset used inverse probability weights (IPW) estimating the average treatment effect (ATE). IPW were computed based on age at 1st HBV testing, sex, birth group, ethnicity, social deprivation quintiles, HCV infection, HIV infection, diabetes, chronic kidney disease, history of injection drug use, alcohol use disorder, and non-alcoholic fatty liver disease. Other ***: White, Black, Central Asian, Latin American, Pacific Islander, and West Asian individuals.

**Table 3 viruses-14-02579-t003:** Impact of HBV infection, NAFLD, and HCV on liver-related mortality between 1990–2015 in the British Columbia Hepatitis Testers Cohort, using the multivariable* Fine–Gray sub-distribution proportional hazards model analysis.

	Overall Study Population		IPW-Dataset ** ATE	
Covariates	asHR	95%CI	asHR	95%CI
**HBV and NAFL status**				
HBV Negative without NAFLD	Ref		Ref	
HBV Negative with NAFLD	4.49	(3.35, 6.02)	4.52	(3.33, 6.12)
HBV positive without NAFLD	3.35	(3.16, 3.55)	3.32	(3.13, 3.53)
HBV positive with NAFLD	12.5	(7.08, 22.07)	12.08	(5.87, 24.89)
**HBV and HCV status**				
HBV Negative without HCV	Ref			
HBV Negative with HCV	4.69	(4.37, 5.03)	4.82	(4.48, 5.19)
HBV positive without HCV	4.36	(4.09, 4.64)	4.3	(4.04, 4.57)
HBV positive with HCV	8.4	(7.62, 9.26)	8.66	(7.77, 9.64)

The impact of HBV infection and NAFLD was assessed using * Multivariable models and these were adjusted with sex, age at first HBV testing date, birth cohort, ethnicity, material deprivation quintiles, HCV infection, diabetes, chronic kidney diseases, alcohol use disorder, injection drug use, and other competing risks. ** The PS-weighted dataset used inverse probability weights (IPW) estimating the average treatment effect (ATE). IPW were computed based on age at the first HBV testing, sex, birth group, ethnicity, social deprivation quintiles, HCV infection, HIV infection, diabetes, chronic kidney disease, history of injection drug use, and alcohol use disorder. The impact of HBV infection and HCV coinfection was assessed using * Multivariable models and these were adjusted with sex, age at first HBV testing date, birth cohort, ethnicity, material deprivation quintiles, diabetes, chronic kidney diseases, alcohol use disorder, injection drug use, and other competing risks. ** The PS-weighted dataset used inverse probability weights (IPW) estimating the average treatment effect (ATE). IPW were computed based on age at 1st HBV testing, sex, birth group, ethnicity, social deprivation quintiles, HIV infection, diabetes, chronic kidney disease, history of injection drug use, alcohol use disorder, and non-alcoholic fatty liver disease. asHR, adjusted sub distributional hazard ratios; CI, confidence interval; HBV, hepatitis B virus, HCV, hepatitis C infection; NAFLD, non-alcoholic fatty liver diseases; IPW, inverse probability weighting; PS, propensity score.

## Data Availability

The data that support the findings of this study are available upon request from the corresponding author. The data are not publicly available due to privacy or ethical restrictions”. Requests to access these datasets should be directed to naveed.janjua@bccdc.ca.

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
