# Peer review of "Impact of Hepatitis B Virus Infection, Non-alcoholic Fatty Liver Disease, and Hepatitis C Virus Co-infection on Liver-Related Death among People Tested for Hepatitis B Virus in British Columbia: Results from a Large Longitudinal Population-Based Cohort Study"

_viruses, 2022, doi:10.3390/v14112579_

Round 1

Reviewer 1 Report

Makuza and colleagues investigated the concurrent impact of HBV infection, NAFLD and HCV coinfection on liver-related deaths in British Columbia (BC), Canada, based on a large population-based cohort study. Their findings provided evidence that HBV infection was associated with a higher risk of liver-related mortality, with a greater relative impact in people with NAFLD and those HCV coinfected. Overall, this manuscript was in general well organized. There are also a few concerns that will need to be addressed.

1) More information is suggested to be provided in the Introduction, especially on the relationship between HBV infection, HCV infection or the NAFLD alone and liver-related mortality, and relationship between HBV-HCV dual infection and liver-related mortality, which, actually, has been reported in several studies.

2) Detailed description and brief summary of the data are suggested to be added in the Result part, such as the presentation of the data in Table 2 on the other factors associated with increased incidence of liver related deaths and the major conclusions that can be drawn from Table 2 and Table 3.

3) The quality and size of Figure 2A and 2C should be improved and optimized.

4) lines 249-250, this sentence should be revised.

Author Response

Comments to the Author

Makuza and colleagues investigated the concurrent impact of HBV infection, NAFLD, and HCV coinfection on liver-related deaths in British Columbia (BC), Canada, based on a large population-based cohort study. Their findings provided evidence that HBV infection was associated with a higher risk of liver-related mortality, with a greater relative impact in people with NAFLD and those HCV coinfected. Overall, this manuscript was in general well organized. There are also a few concerns that will need to be addressed.

  1. More information is suggested to be provided in the Introduction, especially on the relationship between HBV infection, HCV infection, or the NAFLD alone and liver-related mortality, and relationship between HBV-HCV dual infection and liver-related mortality, which, actually, has been reported in several studies.

Responses: Thank you for the comments. We addressed and added more information on HBV, HCV and NAFLD alone as well as coinfection HBV/HCV on liver-related mortality in the updated manuscript introduction.

 “Findings from a study conducted in the United States showed that in 2020, people with chronic HBV infection had 13.3 folds increased risk of dying from liver-related mortality compared to people with no HBV infection (16). People with chronic HCV infection are at higher risk of dying from liver-related death compared to people without HCV infection(17). Treatment of HCV infection with direct-acting antiviral agents (DAAs), is associated with reduction in liver-related mortality (6,18). Several studies have shown that the risk of liver-related mortality is greater for individuals with co-infection HBV and HCV than those mono-infected patients(15,19,20). Literature has shown the high risk of liver-related mortality among people with NAFLD compared to the general population(21).

  1. Detailed description and brief summary of the data are suggested to be added in the Result part, such as the presentation of the data in Table 2 on the other factors associated with increased incidence of liver related deaths and the major conclusions that can be drawn from Table 2 and Table 3.

Responses: Thank you for the comments. We have provided description of results from table 2 and 3 in the manuscript as following.

 “Other factors associated with increased incidence of liver-related deaths include older age at HBV testing (asHR=10.32; 95%CI: 8.20, 13.0 for 65 years old and above compared to individuals of 25 years and below), diabetes (aHR=1.68; 95%CI: 1.59, 1.77), HCV infection(asHR=3.68; 95%CI: 3.45, 3.93), NAFLD(asHR=4.31; 95%CI: 3.32, 5.61), and alcohol use disorder(asHR=3.58; 95%CI: 3.38, 3.80).”

  1. The quality and size of Figure 2A and 2C should be improved and optimized. Responses: Response: Thank you for the comments. In the updated document, we improved the quality and the size of figures 2A and 2C.
  2. In lines 249-250, this sentence should be revised.

Response: Thank you for the comments, this was a typo error, the lacked verb in this sentence was added in the main text in the following sentence.

“These findings highlight that people 

with HBV infection who have concurrent NAFLD, or HCV infection are at much…”

Reviewer 2 Report

The authors have adequately addessed the pros and cons of the study the discussion. These kind of poulation based studies are much needed to implement control programs.

 The study is based on a well-established relationship between Viral diseases associated with liver and metabolic diseases like NFLAD. Still study based poulation provides trends of disease patterns which seem to evolve from time to time. Even though the authors have provided extensive discussion about the strengths and weaknesses of their study it would be helpful if the authors are able to address the following:

  1. Hepatitis vaccination is an effective strategy to prevent the disease, if the authors could cite articles, where the impact of vaccination status was studied, will help to provide more depth to the discussion part.
  2. There are many other population-based, and invitro studies connecting NFLAD and viral infections that are available which could be cited in the discussion section.
  3. Do you have any additional information or data regarding the BMI, and overall lifestyle status of the population under study as it is important to tease out the risk factors involved with NFLAD in the study population?
  4. With regarding the hepatitis C infected group, do you have relevant data in the population regarding recent blood transfusions or any other related risk factor which could bring out a better picture of the disease pattern in the population?

Author Response

The study is based on a well-established relationship between Viral diseases associated with liver and metabolic diseases like NFLAD. Still study-based population provides trends of disease patterns, which seem to evolve from time to time. Even though the authors have provided extensive discussion about the strengths and weaknesses of their study it would be helpful if the authors are able to address the following:

  1. Hepatitis vaccination is an effective strategy to prevent the disease, if the authors could cite articles, where the impact of vaccination status was studied, will help to provide more depth to the discussion part.

Responses: Thank you for the comments. The current paper looked at the mortality associated with HBV, and concurrent impacts of HCV and NAFLD. Vaccination is an important tool to prevent infection and hence disease outcomes such as liver disease and mortality. Since we were not assessing the impact of vaccination as well as treatments, we have included these in the conclusion/recommendation section.

  1. There are many other population-based, and invitro studies connecting NFLAD and viral infections that are available which could be cited in the discussion section.

Responses: We appreciate the suggestion from the review on including other population-based and in vitro studies connecting NAFLD and viral infection, but most of these studies assessed the relationship between NAFLD and viral infections, which is different from our goal of evaluating of the concurrent impact of NAFLD and viral infections on liver-related mortality. We have included data on impact of concurrent viral infection and NAFLD.

  1. Do you have any additional information or data regarding the BMI, and overall lifestyle status of the population under study as it is important to tease out the risk factors involved with NFLAD in the study population?

Responses: We appreciate the suggestion from the reviewer. Unfortunately, we do not have data on BMI and other lifestyle factors in our study. We have added this to the study’s limitations.

  1. With regarding the hepatitis C infected group, do you have relevant data in the population regarding recent blood transfusions or any other related risk factor, which could bring out a better picture of the disease pattern in the population?

Responses: We appreciate the suggestion from the reviewer. We don’t have access to lifestyle and other risk factor for viral infection transmission such as blood transfusion. However, we have access to some risk factor data such as injection drug use, mental illness, and problematic alcohol use, which are independently associated with outcomes and were evaluated for their association in our analyses.

Reviewer 3 Report

General: The authors provide a sorely needed North America population-based (outside of Alaska) examination of hepatitis B mortality, as well as the heightened risk of LRM when CHB occurs in the presence of NAFLD and HCV coinfection. This work clearly illustrates and quantifies the deleterious effects of these comorbidities. I was a little disappointed that age at HBV death was not investigated, as premature mortality is certainly a feature of persons dying of HBV in the US, and I was especially disappointed that antiviral treatment data were not incorporated into this analysis, as it seems those data are available to the investigators. As it stands, HBV antiviral treatment expansion seems critical, as there is indirect evidence that NA-treated pts have markedly improved survival, yet AASLD and other societies have been slow to simplify and liberalize HBV tx guidelines, relying on long-term prospective studies that take years to finish and provide evidence only for incremental changes in their byzantine treatment recs. Additional comments are below.

Abstract: No specific comments.

Background: Although this group has published extensively, there are likely readers (me included) who could benefit from more background on hepatitis B epidemiology in BC. What is acute B incidence? What fraction of CHB pts are Canadian vs non-Canadian-born? Are most chronic cases in urban areas where Asian-born persons live? Is there transmission and how prevalent are CHB cases among indigenous populations in BC?

Methods: In Section 2.2, can the authors elaborate on what they mean by “HBV infection” (which is ultimately revealed in the limitations) and the BC testing guidelines (at least briefly)? Also, it is not revealed until the limitations how multiple cause of death data were used (underlying vs contributing, etc.), and it is not clear to me how they classified an “HBV death.” Could hep B be listed as an underlying or contributing COD? The implication seems yes, which is correct.

In Section 2.3, I think stratification by foreign vs Canadian birth could be important…it is in the US. FB typically have a few more decades of chronic HBV infection than native-born persons typically infected in adolescence or adulthood, so conceivably a mortality difference. The authors mention that 4 ethnicity categories were used but in the tables I don’t see White and it is not clear who “Others” represent – please elaborate/clarify.

Section 2.4, line 105, “crude mortality rate” should include “liver-related,” unless all cause was intended. Line 117, should be “confounded” instead of confounding, as written?

Results: In Section 3.1, line 145, it appears a sentence begins with numeric data. In lines 150-154ish, the distinction between material and social deprivation may not be apparent immediately to most readers (and it’s not explained in the Methods), and for me it was compounded by the seeming contradiction of data for social vs material vulnerability, relative to those most vs least deprived. Please clarify.

Discussion: As mentioned, some of the limitations provided near the end of the paper would have been good to know earlier, in the Methods. I wondered about how HBV deaths were defined from MCOD data, and I assumed/wondered whether the term “HBV infection” meant chronic. In line 312, I would change “could’ to “should.” I was disappointed that no treatment data were worked into this analysis when it appears this information is available – perhaps it is intended for a future paper. It is critical to expand antiviral eligibility, and some data from Asia suggest the risk of death of treated CHB patients is similar to those without HBV infection.

Tables/Figures: Table 1, only 3 ethnicity categories are used – who are “Other?” There also appears to be risk factor data included for which the source is unclear – I did not see in the Methods anything about how IDU, MSM, etc. data were collected (sorry if I missed it). Table 2 includes an aOR of >2 for “Other” vs other ethnicities? Who are these people? Consistency in ethnicity classification for all tables seems needed.

Author Response

General: The authors provide a sorely needed North America population-based (outside of Alaska) examination of hepatitis B mortality, as well as the heightened risk of LRM when CHB occurs in the presence of NAFLD and HCV coinfection. This work clearly illustrates and quantifies the deleterious effects of these comorbidities. I was a little disappointed that age at HBV death was not investigated, as premature mortality is certainly a feature of persons dying of HBV in the US, and I was especially disappointed that antiviral treatment data were not incorporated into this analysis, as it seems those data are available to the investigators. As it stands, HBV antiviral treatment expansion seems critical, as there is indirect evidence that NA-treated pts have markedly improved survival, yet AASLD and other societies have been slow to simplify and liberalize HBV tx guidelines, relying on long-term prospective studies that take years to finish and provide evidence only for incremental changes in their byzantine treatment recs. Additional comments are below.

Responses: Thank you for the comments. The current analysis focused on impact of concurrent HBV, HCV and NFLD and mortality. Thus, we did not include additional data on treatment impact. This analysis sets the stage for future analyses investigating impact of treatment of HBV or HCV among people with concurrent conditions and its impact on outcomes. However, we did not look at age effect, as this is important while looking at mortality. In our further analyses, we will explore age at death to understand impact of HBV on premature deaths. We have previously assessed impact of HCV and substance use on mortality and age at death.   

  1. Background: Although this group has published extensively, there are likely readers (me included) who could benefit from more background on hepatitis B epidemiology in BC. What is acute B incidence? What fraction of CHB pts are Canadian vs non-Canadian-born? Are most chronic cases in urban areas where Asian-born persons live? Is there transmission and how prevalent are CHB cases among indigenous populations in BC?

Response: Thank you for the suggestion, available information on HBV epidemiology in BC and Canada in general was added in the main text as you can see it in the following sentences, however all suggested information were not retrieved.

Immigrants from East Asia (5%-15%) have a higher prevalence then the Canadian-born population (0.24%-0.5%)(9). In British Columbia (BC), among people diagnosed with HBV infection, 4.3% had acute HBV infection while 95.7% had a chronic infection (10). In addition, acute HBV infection was predominantly diagnosed among White people (78%), while 60% of chronic HBV were among people of East Asian origin (10).

  1. Methods: In Section 2.2, can the authors elaborate on what they mean by “HBV infection” (which is ultimately revealed in the limitations) and the BC testing guidelines (at least briefly)? Also, it is not revealed until the limitations how multiple cause of death data were used (underlying vs contributing, etc.), and it is not clear to me how they classified an “HBV death.” Could help B be listed as an underlying or contributing COD? The implication seems yes, which is correct.

Response: - Thank you for the suggestion, we added the definition of HBV infection in the section 2.2 in the main text as you could see it in the following sentence.

In the BC HTC, an individual was confirmed as a case of HBV infection with a positive  HBV DNA or HBVe antigen (HBeAg) test result, or those who had a record of receiving HBV treatment (13).

  • As our outcome was liver-related mortality, we did not define HBV alone; we considered all liver-related mortality causes, including death from viral hepatitis (ICD10, B15-19) where HBV belongs.

  1. In Section 2.3, I think stratification by foreign vs Canadian birth could be important…it is in the US. FB typically have a few more decades of chronic HBV infection than native-born persons typically infected in adolescence or adulthood, so conceivably a mortality difference. The authors mention that 4 ethnicity categories were used but in the tables I don’t see White and it is not clear who “Others” represent – please elaborate/clarify.

Response: - Thank you for the suggestion. This is an important aspect to consider. However, within the BC-HTC, ethnicity is derived based on name recognition and not based on immigration data. Thus, it is difficult to differentiate between Canadian born vs foreign born individuals. We certainly do these assessments when immigration data become available.

  • The 4 ethnicity categories used was an error, which is now rectified. In addition, we have provided clarification of group included in the other category in the main test and in the tables’ legends.
  1. Section 2.4, line 105, “crude mortality rate” should include “liver-related,” unless all cause was intended. Line 117, should be “confounded” instead of confounding, as written?

Response: Thank you for the comment. We have rectified this error as below.

We assessed the crude liver-related mortality rate…”, “…was whether the covariates confounded the relationship between HBV…

  1. Results: In Section 3.1, line 145, it appears a sentence begins with numeric data. In lines 150-154ish, the distinction between material and social deprivation may not be apparent immediately to most readers (and it’s not explained in the Methods), and for me it was compounded by the seeming contradiction of data for social vs material vulnerability, relative to those most vs least deprived. Please clarify.

Response: - Thank you for the comments. We have revised sentence in line 145 as:

“In total 8,963(0.81%) individuals…”

  • For line 150-154, we have provided definition of material and social deprivation in the methods as: “The material component consisted of indicators for education, employment, and income (persons without high school diplomas, ratio employment-population ratio, and average personal income). In contrast, the social component comprised indicators related to marital status and family structure (persons living alone, persons separated, divorced or widowed, and single-parent families)(10)
  1. Discussion: As mentioned, some of the limitations provided near the end of the paper would have been good to know earlier, in the Methods. I wondered about how HBV deaths were defined from MCOD data, and I assumed/wondered whether the term “HBV infection” meant chronic. In line 312, I would change “could’ to “should.” I was disappointed that no treatment data were worked into this analysis when it appears this information is available – perhaps it is intended for a future paper. It is critical to expand antiviral eligibility, and some data from Asia suggest the risk of death of treated CHB patients is similar to those without HBV infection.
  2. Response: Thank you for the suggestion.

- We have provided important methods and cohort details in the methodology section and data limitation, which were important for interpretation of findings, are provided in the limitation section.

- The term “HBV infection” mean chronic HBV infection.

- Treatment was not included in this analysis. However, treatment data are being incorporated in our upcoming papers, which are in progress.

  1. Tables/Figures: Table 1, only 3 ethnicity categories are used – who are “Other?” There also appears to be risk factor data included for which the source is unclear – I did not see in the Methods anything about how IDU, MSM, etc. data were collected (sorry if I missed it). Table 2 includes an aOR of >2 for “Other” vs other ethnicities? Who are these people? Consistency in ethnicity classification for all tables seems needed.

Response: - Thank you for the comments. We have corrected this error. “Other” in ethnicity was explained in method as well as in Table1 and table 2 in in the main text. It include White, Black, Central Asian, Latin American, Pacific Islander, and West Asian individuals.

“ ..Other: White, Black, Central Asian, Latin American, Pacific Islander, and West Asian individuals).

  • In the method, description for assessment of IDU, MSM, and other variables has been added as: “We defined as the history of injection drug use (IDU), gbMSM, and problematic alcohol use at the time of the last HBV diagnosis or the last negative test (for negative individuals) using validated algorithms and determined at the occurrence of either one hospitalization or ambulatory care visit, or two physician visits pertaining to these risk factors(12,13)”.